# Cognitive and Interpersonal Factors in Adolescent Inpatients with Anorexia Nervosa: A Network Analysis

**DOI:** 10.3390/children10040730

**Published:** 2023-04-15

**Authors:** Chantal P. Delaquis, Nathalie T. Godart, Melina Fatséas, Sylvie Berthoz

**Affiliations:** 1INCIA CNRS UMR 5287, Université de Bordeaux, 33000 Bordeaux, France; 2Fondation Santé des Etudiants de France, 75014 Paris, France; 3CESP, University Paris-Sud, UVSQ, INSERM U 1178, Université Paris-Saclay, 94805 Villejuif, France; 4UFR Simone Veil-Santé, Université Versailles Saint-Quentin-en-Yvelines, 78047 Montigny-le-Bretonneux, France; 5Department of Addictology, CHU Bordeaux, 33000 Bordeaux, France; 6Department of Psychiatry, Institut Mutualiste Montsouris, 75014 Paris, France

**Keywords:** anorexia, adolescent, network analysis, cognitive-interpersonal model, perfectionism, alexithymia

## Abstract

The cognitive-interpersonal model of anorexia nervosa (AN) posits that cognitive and interpersonal traits contribute to the development and maintenance of AN. We investigated cognitive and interpersonal factors put forward by the model in a sample of 145 adolescent inpatients with AN using network analysis. Our main outcomes included core eating disorder symptoms, cognitive style, socio-affective factors, and mood symptoms. We estimated a cross-sectional network using graphical LASSO. Core and bridge symptoms were identified using strength centrality. Goldbricker was used to reduce topological overlap. The node with the highest strength centrality was Concern over Mistakes, followed by Eating Preoccupation, Social Fear, and Overvaluation of Weight and Shape. The nodes with the highest bridge strength were Concern over Mistakes, Doubt about Actions, Overvaluation of Weight and Shape, and Depression. Notably, both performance on a cognitive flexibility task and BMI were not connected to any other nodes and were subsequently removed from the final network. We provide partial support for the cognitive-interpersonal model while also supporting certain premises put forward by the transdiagnostic cognitive-behavioral model. The high centrality of Concern over Mistakes and Social Fear supports the theory that both cognitive and interpersonal difficulties contribute to AN, particularly in adolescence.

## 1. Introduction

Anorexia nervosa (AN) is a serious psychiatric disorder characterized by a fear of weight gain, distorted body image, and extreme weight-control behaviors [1]. AN is associated with socio-affective difficulties [2], psychological comorbidity, medical complications, and high mortality [3]. Nearly all cases have their onset before the age of 25 years [3,4]. Given that AN has its peak onset in adolescence, it is critical to understand the developmental and maintenance factors during this period.

### 1.1. The Cognitive-Interpersonal Model of AN

The cognitive-interpersonal model of AN, proposed in the mid-2000s, posits that obsessive-compulsive (OC) and avoidant traits are predisposing factors to the development of AN that also maintain the disorder by fostering pro-anorexia beliefs and behaviors [5,6]. For example, high trait perfectionism may lead to the development of strict personal standards about one’s weight, as well as rigid and extreme behaviors to prevent weight gain. Perfectionistic and avoidant traits may also contribute to interpersonal difficulties, and the emotional reactions of others to extreme weight loss are considered to be a maintaining factor. The cognitive-interpersonal model argues that those with AN stand out from other people with EDs due to the highly visible nature of the disorder, which arouses strong emotions in others. Despite these negative reactions, many people with AN appear oblivious to the dangers of their illness and staunchly defend their extreme dietary restraint, putting strain on interpersonal relationships.

A theory at the forefront of eating disorder (ED) research is the transdiagnostic cognitive-behavioral model, proposed several years before the cognitive-interpersonal model [7]. In contrast to the transdiagnostic model, the cognitive-interpersonal model suggests that there is a unique and separate phenotype that characterizes AN, particularly the restricting subtype (AN-R) [5]. Further, a core premise of the transdiagnostic cognitive-behavioral model is that overvaluation of weight and shape is central to all EDs. This refers to the dysfunctional self-evaluation system in which self-worth is based mainly on one’s weight, body shape, and eating habits, regardless of achievements and qualities in other areas of life [7]. In contrast, the cognitive-interpersonal model considers dietary restraint to be the essence of AN. Weight and shape concerns may be one motivation for restraint, but they are not central. Restraint is seen as a defense mechanism against strong emotions. As starvation leads to intense preoccupation with food, the person with AN must stay vigilant to not give in to the urge to eat. As a consequence of being constantly focused on food and eating, other emotions are thought to become less salient, and many people with AN report feeling emotionally numb or lacking emotional clarity [5].

#### 1.1.1. Cognitive Profile

Obsessive-compulsive disorders and EDs are highly comorbid, particularly in AN [8]. Cognitive processing styles, such as perfectionism, lack of cognitive flexibility (i.e., set shifting), and meticulous attention to detail (i.e., low central coherence), are thought to be connected to dietary restraint.

Perfectionism is a personality trait that reflects the tendency to have very high personal standards, be overly critical of oneself when these standards are not achieved, have an extreme preoccupation with mistakes, doubt when considering personal achievements, and have an excessive focus on organization and precision [9]. Higher levels of perfectionism are consistently found in adolescents and adults with AN compared to healthy controls [10,11,12] and may be a risk factor for longer illness duration [13]. Perfectionism has also been associated with feeling the need for control over the self and body [14]. In AN, eating anything outside a very limited range of “safe foods” is considered a failure. This black-and-white thinking style leads to the belief that eating must be rigidly controlled, and any deviation from strict dietary rules is considered a failure.

Cognitive flexibility is the ability to adaptively shift between mental processes in response to environmental demands and make subsequent appropriate behavioral adjustments. Poor cognitive flexibility tends to go hand in hand with difficulty balancing detail-oriented and global information processes (i.e., weak central coherence) [15]. Lack of cognitive flexibility is thought to contribute to rigid thoughts and behaviors, making many people intolerant to change, treatment resistant, and prone to relapse [6]. Evidence has shown that adults with AN have impaired cognitive flexibility with attenuated difficulties that persist after recovery [6,16,17,18]. In children and adolescents, evidence is mixed. Meta-analytic evidence found that set-shifting deficits are less pronounced in children and adolescents aged 10–18 compared to adults [19,20], a result the authors interpreted as suggesting that set-shifting difficulties may develop as a result of prolonged starvation. However, another study found no significant difference in set-shifting deficits between adults and adolescents with AN [15]. More research is therefore needed on cognitive flexibility in young people with AN.

#### 1.1.2. Interpersonal Difficulties

Difficulty with interpersonal relationships is a well-established symptom of AN and is thought to be a maintaining factor of the disorder (for review, see [21]). According to the cognitive-interpersonal model, family members and close others may unintentionally reinforce AN by complimenting initial weight loss in the early stages of the illness. As the person’s state of starvation becomes more visible, close others may begin to respond more negatively. This may manifest as frustration at the person’s refusal to “just eat” or enabling AN by organizing meals around dietary rules. Social interactions become increasingly threatening and negative to the person with AN. For instance, social events where food and drink are being consumed may be avoided due to anxiety about eating in public and others’ negative emotions when faced with their weight loss or refusal to eat. Therefore, interpersonal relationships are avoided as they are increasingly characterized by conflict, criticism, and negative emotion.

Evidence suggests that interpersonal difficulties in AN are linked to alexithymia [2,21], defined as difficulty with identifying, analyzing, imagining, and communicating emotions [22]. Alexithymia can be viewed as a deficit in emotion regulation that substantially impacts interpersonal relationships [2]. Social anxiety (SA) and avoidance have both been associated with alexithymia in AN, even while adjusting for comorbid anxiety, depression, and a state of starvation [2]. Further, alexithymia has been associated with deficits in labeling emotional facial expressions [23], which may contribute to difficulties with social cognition. While high levels of alexithymia are found across eating disorders, evidence suggests that those with AN, and particularly the AN restricting subtype (AN-R), may have elevated difficulties with describing emotions compared to bulimia nervosa (BN) [24].

In addition, individuals with AN report high rates of psychiatric comorbidity. More than 50% of adolescents with AN have at least one psychiatric comorbidity, with depression and anxiety being the most commonly reported [25,26]. These comorbid conditions have been associated with increased ED symptoms, hospitalizations, and suicide attempts in people with AN compared to those without depressive or anxious comorbidity [25]. Coexisting psychiatric symptoms have a complex relationship with ED symptomatology, as they may precede, aggravate, or emerge from existing vulnerabilities. For instance, SA often leads to increased social avoidance and isolation, and feelings of isolation and alexithymia are highly associated with depressive mood [27]. These conditions may be further exacerbated by starvation, creating a vicious cycle. Therefore, it is important to investigate the role of general psychopathology symptoms, particularly depressive and anxious symptoms, in EDs in addition to core ED symptoms.

### 1.2. Using Network Analysis to Test Theoretical Models

Network analysis (NA) has been increasingly applied to the study of EDs to model the connections between symptoms and overall network structure [28,29,30]. Network theory proposes that, rather than stemming from an underlying disorder, symptoms directly influence one another in the development and maintenance of psychopathology. In psychological networks, nodes represent symptoms and edges represent the connections between nodes [28]. NA can identify potential causal pathways between symptoms, central symptoms, and clusters that are more tightly connected than others [30]. Central nodes are nodes that are highly connected and thus important in maintaining the network, indicating potential treatment targets. In networks containing symptoms from two or more disorders, bridge nodes indicate which symptoms connect the two communities to spread activation from one disorder to another [31]. NA is particularly suited to test theoretical models, as a network of symptoms can be directly compared to the theorized relationships between symptoms [30].

NA has been used in several studies to investigate the transdiagnostic cognitive-behavioral model of eating disorders [32,33,34,35,36]. While these studies have consistently found overvaluation of weight and shape to be a central symptom across eating disorders, several limitations have been noted. First, the majority of these studies used questionnaire items as nodes rather than subscale scores, as recommended [29]. While using questionnaire items can be appropriate for certain research questions, including multiple items from the same scale increases the risk of topological overlap between nodes, inflating the centrality of certain nodes and changing network structure [37]. In addition, although no sample size guides exist for NA, a general recommendation is to estimate parameters based on the sample size to ensure appropriate network stability; therefore, when investigating many variables, an item-based analysis may not be feasible [38].

Although both the cognitive-behavioral and cognitive-interpersonal models posit that people with EDs experience a wide range of symptoms, including perfectionism and interpersonal difficulties, the majority of network models examine only core ED symptoms. Networks are highly sensitive to included variables, and the inclusion of general psychopathology symptoms in addition to core ED symptoms may affect the centrality of the results [37,39]. More recently, NAs that investigate a broader range of symptoms in transdiagnostic, mixed samples of adults and adolescents have found that perfectionism [14,40], depression, and interpersonal difficulties [41,42] all emerge as central symptoms. In a transdiagnostic sample of adolescents aged 10–15 years old, bridge nodes between ED symptoms and depression included irritability, social eating, and depressed mood [43]. Investigating comorbid symptoms in people with ED is particularly relevant in adolescence, as general psychopathology, including EDs, tends to emerge during this developmental period [44].

Most network studies examine mixed-age and transdiagnostic samples, with a few exceptions. One NA comparing ED symptoms in adolescent and adult patients with AN and BN found similarities in network structure across ages and diagnoses, showing that drive for thinness, ineffectiveness, perfectionism, and emotion dysregulation are potentially important treatment targets [45]. Another NA examined ED symptoms across five developmental stages and found that while ED symptoms such as food avoidance and overvaluation of weight and shape emerged as central, the interconnectivity of the network increased with age [46]. As non-ED symptoms such as depression, social anxiety, and interpersonal difficulties appear to be central in both adult and adolescent networks, there is a need for more research on the relationship between general psychopathology and core ED symptoms in specific populations.

### 1.3. The Current Study

We aim to investigate cognitive and interpersonal factors posited by the cognitive-interpersonal model using NA in a sample of adolescent inpatients with AN. Our definition of adolescence in the current analysis is in line with recent evidence indicating that biological and social adolescent growth correspond more closely with the period between 10 and 24 years old, rather than 10–19 years old [47]. We hypothesize that restraint will be more central than overvaluation of weight and shape and that nodes related to cognitive and interpersonal symptoms (e.g., perfectionism, family dynamic) will emerge as important bridge nodes to core ED symptoms.

## 2. Materials and Methods

### 2.1. Study Population

This study is part of a longitudinal multicentred trial called EVHAN (Evaluation of Hospitalization for AN). A subset of 145 inpatients with AN was included in the present analysis. Inclusion criteria for the current study were as follows: being hospitalized for AN due to a high risk to life, either physical or psychological (i.e., suicidality), having an admission BMI < 15 and/or sudden and rapid weight loss, consent for participation in the study, and affiliation with the French Social Security health coverage system. Exclusion criteria were refusal to participate, insufficient command of the French language, and presence of a potentially confounding medical pathology (e.g., diabetes, Crohn’s disease, celiac disease, and other metabolic disorders). For the present analysis, two additional criteria were added. First, participants had to be aged between 13 and 24 years, as patients younger than 13 years were not assessed with the same test battery. Next, participants must have completed the Test of Attentional Performance (TAP)—Flexibility subtest.

Participants were recruited from 11 inpatient eating disorder treatment facilities in France (CHU Bordeaux, Cochin—Maison des Adolescents, Institut Mutualiste Montsouris, MGEN—La Verrière, CHU-Nantes, CHU-Rouen, Robert Debre Hospital, Sainte-Anne Hospital, Saint-Etienne Hospital, Villejuif—Paul Brousse) between April 2009 and May 2011. This research was approved by the Comité de Protection des Personnes Ile-de-France III on 25 March 2008.

### 2.2. Measures

Data were collected within two weeks of the patient’s hospital admission for AN, including sociodemographic data, age of AN onset, self-report questionnaires, neuropsychological evaluation, and nutritional status. Patients were diagnosed with current AN based on DSM-IV-TR criteria. The BMI criterion was <10th percentile up to 17 years of age and BMI < 17.5 for 17 years of age and above. For all self-report measures, subscale or total scores were included in the network. Including single items would have led to a large number of nodes and thus reduced the accuracy of our network estimation given the present sample size [38]. More information on node selection is presented below.

#### 2.2.1. Self-Report Questionnaires

Eating Disorder Examination Questionnaire (EDEQ). Eating disorder symptoms were evaluated using a French translation of the Eating Disorder Examination Questionnaire (EDEQ; [48,49]), a 28-item questionnaire that assessed specific eating pathology in the last 28 days, with 22 items rated on a scale from 0 to 6. An additional six open-ended items assessed the frequency of binge eating, vomiting/laxative abuse, and excessive exercise. The EDEQ has four subscales: Restraint, Eating Concern, Weight Concern, and Shape Concern. The Restraint subscale includes items related to avoidance of food and eating, dietary rules, and desire for an empty stomach. Eating Concern measures preoccupation with food, guilt about eating, fear of losing control over eating, and secret and social eating. Shape Concern includes feelings of fatness, discomfort with shape and body, fear of weight gain, and preoccupation with shape. Weight Concern measures the importance, dissatisfaction, and preoccupation with weight as well as the desire to lose weight. The four subscales were included in the present analysis.

Frost Multidimensional Perfectionism Scale (FMPS). The FMPS [9] (French version [50]) evaluates perfectionism with 35 items grouped into six subscales: Concern over Mistakes, Personal Standards, Parental Expectations, Parental Criticism, Doubt about Actions, and Organization. Concern Over Mistakes reflects negative reactions to mistakes, interpreting mistakes as failure, and the belief that one will lose the respect of others if they make a mistake. Personal Standards refers to the tendency to set very high standards for oneself and place excessive importance on meeting those standards. Parental Expectations refers to the belief that one’s parents have very high standards, and the perception that one’s parents are overly critical is reflected in the Parental Criticism subscale. Doubt about Actions consists of items from the Maudsley Obsessive-Compulsive Inventory and reflects the extent to which people doubt their ability to accomplish tasks. Organization refers to the tendency to value neatness and organization. Items are rated on a five-point Likert scale and demonstrate sound psychometric properties in patients with AN [51].

**Bermond–Vorst Alexithymia Questionnaire (BVAQ).** The BVAQ measures alexithymia, defined as a reduction or incapacity to experience, verbalize, fantasize, and/or think about one’s emotions [22] (French version [52]). People with AN have been found to have high levels of alexithymia [53], indicating socioemotional difficulties. The BVAQ consists of five subscales, each comprising of eight items: Emotionalizing, Fantasizing, Identifying, Analyzing, and Verbalizing. These subscales can be split into two factors, namely cognitive and affective alexithymia. The cognitive factor is obtained by summing Identifying, Analyzing, and Verbalizing subscales and the affective factor is obtained by summing Emotionalizing and Fantasizing. In the present network, the total alexithymia score was used to reduce topological overlap, increase network stability, and map onto the more commonly used Toronto Alexithymia Scale [54,55], which does not differentiate between cognitive and affective facets.

Liebowitz Social Anxiety Scale (LSAS). The LSAS is a clinician-rated assessment for social phobia to measure the range of social and performance situations that individuals fear and avoid [56,57] (French version [58]). Consisting of 24 items rated on a scale from 0 to 3, it is divided into two subscales: Fear and Avoidance. Of note, in the current study, one hospital center failed to collect data for the avoidance subscale, resulting in missing data for *n* = 17 patients.

Hospital Anxiety and Depression Scale (HAD). The HAD consists of eight items assessing depression and eight items assessing anxiety rated on a five-point (0–4) Likert scale [59] (French version [60]). The HAD determines the presence and severity of the most common depressive and anxiety symptoms. Depression and anxiety scores range from 0 (no symptoms) to 21 (severe symptoms).

Family Assessment Device (FAD). The FAD is designed to assess a number of dimensions of perceived family functioning [61] (French version [62]). This scale includes 12 items that assess the overall health/pathology of the family on a scale that includes four possible responses: strongly agree, agree, disagree, and strongly disagree. Items include we feel accepted for what we are, we confide in each other, and we avoid discussing our fears and concerns. Items are reverse coded if necessary, and a total score is calculated from 1 to 4, with 1 reflecting healthy functioning and 4 reflecting unhealthy functioning.

#### 2.2.2. Neuropsychological Evaluation

Test of Attentional Performance 2.1—Flexibility. The FLEX is a subtest of the Test of Attentional Performance version TAP 2.1 [63]. The FLEX is a set-shifting task. The “non-verbal” condition was chosen to eliminate the potential bias of age-related differences in verbal ability. In this condition, angular and rounded figures (Figure 1) are presented simultaneously on the screen. The subject must indicate, as quickly as possible, using the left or the right response button, depending on whether the target stimulus appears to the right or left of the center of the screen. The instruction is to respond alternately to the angular and then to the rounded shapes, starting with the angular shape. The test consists of 100 trials. Unfortunately, we were unable to calculate the flexibility index for our sample, as there are no established population norms in adolescents for this task. Therefore, performance was measured by the median reaction time for the correct answers.

#### 2.2.3. Nutritional Status

Body Mass Index (BMI). Body weight and height were measured by using, respectively, a standard tilt scale (Omega-SECA, Hamburg, Germany) and a stadiometer (wall-mounted model 222-SECA, Hamburg, Germany) scale. BMI is obtained by dividing weight (kg) by height squared (meters).

### 2.3. Statistical Analysis

Data analysis was conducted in R version 4.2.1 with RStudio version 2022.12.0 + 353. See Appendix A for the code used. The current study follows recently published reporting standards for psychological network analyses [64].

Although general sample size recommendations have been proposed, there exist no definitive guidelines and power analyses for NA. Therefore, it is critical to examine the stability of the network before interpretation. The higher number of parameters that are estimated in the NA, the more participants are required to ensure stability [38]. Missing data were not imputed, and pairwise complete observations were used to handle missing data. After node selection, the final network included 14 nodes: 3 ED, 1 alexithymia, 2 general psychopathology, 5 perfectionism (including 2 parental variables), 1 family dynamic, and 2 social anxiety variables (BMI and FLEX were not included in the final network. See Appendix A for details). We estimated a network using data from all participants to examine node centrality and conducted a bridge node analysis to identify potential nodes bridging ED, interpersonal, and cognitive symptoms. A post hoc exploratory network was conducted to compare AN subtypes.

#### 2.3.1. Data Preparation

We examined data means, standard deviations (SD), skewness, and kurtosis. Three tests of multivariate normality were conducted: Mardia’s test, Henze–Zirkler’s test, and Royston’s test. All three tests indicated that the multivariate normality assumption was violated and that the univariate normality assumption was violated for the majority of outcome variables. Therefore, we used the nonparanormal transformation, which relaxes the normality assumption, with the R-package huge version 1.3.5, as recommended [38].

#### 2.3.2. Node Selection

In network analysis, careful selection of the included variables is critical. The inclusion of multiple nodes that assess the same underlying construct can inflate the centrality of these variables [37]. Variable selection should be data-driven and informed by theory. The goldbricker function in the networktools package detects topological overlap. We searched the database for redundancies with this function with a threshold of 0.20 (*p* = 0.01). If clinically appropriate, nodes with significant overlap were combined using principal component analysis using the net_reduce function. In line with previous literature (e.g., [14,33]), we opted for this method so that our network was correctly represented, although combined nodes may have limited interpretability.

The goldbricker function revealed seven “bad pairs” (see Appendix A). After consultation with clinical experts (SB, NG), we chose two pairs to combine with PCA using the net_reduce function. Here, we provide a list of the combined nodes and a brief justification for their combination based on a clinical perspective and in line with previous literature.
EDEQ Weight Concern and EDEQ Shape Concern: overvaluation of weight and shape, where one’s self-worth is largely determined by body shape and weight. This has been shown to be a central node in several network models of eating disorders (e.g., [32,33,36])FMPS Parental Expectations and FMPS Parental Criticism: both dimensions measure the patient’s perceived parental influence on maladaptive perfectionism


#### 2.3.3. Network Analysis

Network Stability. To estimate edge and centrality stability, we used the bootnet package version 1.5 [38]. Non-parametric bootstrapping (i.e., resampling rows with replacement) was used to create 2500 samples to estimate edge weight stability and difference tests for centrality and bridge measures. Case-dropping bootstrap samples (*n* = 2500) were used to estimate the stability of centrality indices. The correlation stability (CS) coefficient measures the stability of centrality indices by indicating the percentage of our sample that can be dropped to maintain, with a 95% confidence interval, a correlation value equal to or above *r* = 0.7. The CS coefficient is considered acceptable when above 0.25 and good from 0.5 [38].

Network estimation, centrality, and predictability. A regularized partial correlation network was estimated for the entire sample, resulting in an undirected, weighted network. Each edge represents a conditional dependence between two nodes, and an absent edge signifies conditional independence. Edges represent partial correlations. The thickness of the edge represents the strength of the correlation between variables. The network was visualized with the qgraph version 1.9.2 [65], which uses graphical LASSO (least absolute shrinkage and selection operator regularization) with the extended Bayesian information criteria (EBIC) set at a threshold of 0.5 to visualize the networks. The Fruchterman–Reingold algorithm was used for node positioning, where nodes are organized based on the strength of their connections, facilitating visualization.

To inspect centrality measures, we used centralityPlot and centralityTable functions, including the strength (the sum of the absolute values of a node connection) and expected influence (EI; similar to strength except positive and negative values are taken into account), as other centrality indices such as betweenness and closeness have been shown to be less reliable in psychological networks [66]. As we are interested in the relative centrality differences of eating disorder symptoms, the bootstrapped difference test (alpha = 0.05) between Overvaluation of Shape and Weight and both Restraint and Eating were conducted using the bootnet differenceTest function.

Network predictability was assessed with R-package *mgm* version 1.2-13. Predictability refers to how well each node is explained by all other nodes in the network by computing each node’s R^2^ [67].

Bridge nodes. Nodes that connect different clusters of symptoms in psychopathological networks are called bridge nodes. For example, bridge nodes could connect eating disorder symptoms (cluster 1) to depressive symptoms (cluster 2). The bridge function in the package network tools was used to calculate bridge strength (a node’s total connectivity with nodes in other communities in the network) and bridge EI(such as bridge strength but summing positive and negative values) [31].

## 3. Results

### 3.1. Participant Characteristics

Descriptive statistics are presented in Table 1. Participants were *n* = 145 (*n* = 137 female; *n* = 8 male) inpatients diagnosed with AN. The percentage of male participants is consistent with other network ED studies (e.g., [14,35]) and general prevalence estimates [4]. The sample consisted of *n* = 78 (53.8%) participants with AN binge purge subtype (AN-BP) and *n* = 67 (46.2%) with AN-R. Sixteen percent (*n* = 22) of the sample had a premenarchal onset of AN. The average age at hospital admission was 18.2 (SD = 3.1) years old, and *n* = 78 (53.8%) of the sample was under the age of 18. The average age of AN onset was 15.3 (SD = 3.2) years old, and the average duration of illness was 2.9 (SD = 2.5) years.

### 3.2. Network Structure

Table 2 shows all nodes included in the network. The cognitive flexibility (FLEX) outcome and BMI were removed as they were not connected to any other nodes in the network (see Appendix A). The network had acceptable stability, with a CS coefficient of 0.44 for both strength and EI. As all connections in the network were positive, only strength will be discussed. See Figure 2 for the network plot. Mean predictability was 0.45, and the nodes with the highest predictability were Concern over Mistakes (M = 0.66) and Eating (mean = 0.64). The predictability network model is presented in Appendix A.

Figure 3 shows the strength plot (see Appendix A for the bootstrapped difference test). The node with the highest strength centrality was Concern over Mistakes (1.31), followed by Eating (0.99), Fear (0.96), and Overvaluation of Weight and Shape (0.95). The node with the lowest strength centrality was Organization (0.37). The strongest connections in the network were between Eating and Overvaluation of Weight and Shape (part *r* = 0.72), Restraint and Eating (part *r* = 0.67), and Fear and Avoidance (part *r* = 0.63). The strongest connections to the central node Concern over Mistakes were Personal Standards (part *r* = 0.67), Doubt about Actions (part *r* = 0.56), Overvaluation of Weight and Shape (part *r* = 0.5), and Parental Criticism and Expectations (part *r* = 0.49).

Bootstrapped difference tests showed that Overvaluation of Weight and Shape had significantly higher strength centrality than Restraint. There was no significant difference in strength centrality between Overvaluation of Weight and Shape and Eating.

In our post hoc exploratory analysis of AN subtypes, the CS coefficient was <0.25. Therefore, the results of this analysis could not be interpreted and are not included here.

### 3.3. Bridge Nodes

The CS coefficient for bridge strength was 0.44, indicating that centrality indices could be interpreted. As all correlations were positive, only bridge strength is discussed. Bridge strength is shown in Figure 4. The bootstrapped difference test for bridge strength is shown in the Appendix A. The nodes with the highest bridge strength were Concern over Mistakes (0.60), Doubt about Actions (0.42), Overvaluation of Weight and Shape (0.30), and Depression (0.29). Figure 5 depicts the computed bridge clusters. Three clusters were detected. Cluster one consisted of core ED symptoms. Cluster two contained nodes related to negative affectivity, including Depression, Anxiety, and Doubt about Actions. The third cluster grouped cognitive and socio-affective variables, including Alexithymia, Perfectionism, and Family Dynamic.

Regarding bridge pathways, the strongest pathway to core ED symptoms from cluster two was between Depression and Eating (part *r* = 0.4), and from cluster three, the strongest pathway was between Concern over Mistakes and Overvaluation of Weight and Shape (part *r* = 0.5).

## 4. Discussion

The aim of this study was to evaluate the cognitive-interpersonal model in a sample of adolescent inpatients with AN. We investigated the complex relationships between core ED symptoms and cognitive (perfectionism, flexibility), interpersonal (alexithymia, social anxiety, family dynamic), and mood (depression, general anxiety) symptoms. Concern over Mistakes, Preoccupation with Eating, Social Fear, and Overvaluation of Weight and Shape were the most central nodes.

### 4.1. Central Nodes

#### 4.1.1. Core Eating Disorder Symptoms

Eating and Overvaluation of Weight and Shape emerged as the most central ED symptoms. The high centrality of Eating Preoccupation is in line with the cognitive-interpersonal model, which posits that AN patients experience constant thoughts or images related to food and eating as a result of dietary restriction. Our results are also in line with the majority of network studies of both AN and transdiagnostic samples, which indicate that Overvaluation of Weight and Shape is a central ED symptom in both adults and adolescents [33,36,39,46]. Contrary to our hypotheses, Restraint was not a central ED symptom, and it was not connected to cognitive and interpersonal symptoms. This suggests that Overvaluation of Weight and Shape and Eating Preoccupation may play a significant role in severe AN over and above Restraint. Importantly, Eating and Restraint were highly connected. The centrality of Eating suggests that the cognitive consequences of extreme dietary restraint (i.e., preoccupation with food) may be more central than behavioral symptoms (i.e., dietary restraint). As the average duration of illness in our adolescent sample was around three years, this is consistent with evidence that cognitions are more central in the early stages of EDs, but those with a longer duration of illness evolve more deeply engrained behavioral habits that become more central to the pathology [32].

Evidence from previous NAs do suggest that Restraint plays an important role in EDs. In a sample of adult females with ED (54.5% AN-R; 23.4% AN-BP; BN 22.1%), EDEQ Restraint had the highest bridge strength between ED symptoms and nodes related to social cognition [68]. Another study examining mixed self-reported EDs across five developmental stages found that food avoidance was a central symptom regardless of developmental stage [46]. Finally, an NA investigating the progression of the most central symptoms across the duration of illness in the same sample found that fasting was a central symptom in the overall network [32]. In AN specifically (in a sample of mixed full syndrome and subclinical, no subtypes distinguished), one NA investigating the cognitive-behavioral model in adolescents and adults (Mean age = 30) found that EDEQ Restraint was more prominent in AN compared to BN and binge eating disorder [36].

Our finding that Restraint was not a central symptom in an inpatient sample with AN is surprising. At the same time, just because a node is not central does not indicate that it is less clinically important. It is possible that Restraint indirectly contributes to AN by leading to Eating Preoccupation, which subsequently develops into a maladaptive emotion regulation strategy. It is also possible that Restraint connects to other variables not included in our network, such as difficulties in emotion regulation. Future research should investigate the role of Restraint in AN and its potential association with Eating Preoccupation and broader psychopathological variables, such as difficulties in emotion regulation.

#### 4.1.2. Cognitive and Interpersonal Nodes

Concern over Mistakes was the most central symptom in our network. Further, Doubt about Actions was also highly central, in line with evidence that perfectionism plays an important role in AN [10]. One NA of adults and adolescents (mean age = 23 years) with AN found that perfectionistic evaluative concerns (empirically combined Concern over Mistakes with Doubt About Actions) was the most central bridge node to core eating pathology [14]. An NA of a mixed ED sample (42% AN, mean age = 24 years, mean duration of illness = 5.7 years) found that perfectionistic personal standards were highly central [40]. Further, a network study analyzing a subsample of adolescent inpatients with AN also found a high centrality of perfectionism [45]. Perfectionism has been associated with higher anxiety and depression and worse hospitalization outcome (i.e., clinical symptom reduction) in adult (>18 years old) inpatients with AN [69]. Importantly, targeting perfectionism in ED treatment has been shown to lead to reductions in ED symptoms, depression, and anxiety [70]. Therefore, perfectionism appears to be a central symptom in EDs transdiagnostically, and our results indicate that it may have a particular relevance in severe cases of adolescent AN.

Fear related to social anxiety (SA) was also highly central. It has been well established that EDs are highly comorbid with SA, which is associated with more severe ED psychopathology [71]. Further, SA disorder often occurs before ED onset, suggesting that SA is a risk factor for developing an ED [72]. This is in line with the cognitive-interpersonal model, which posits that people with social difficulties have an increased vulnerability to developing AN, and that these social difficulties maintain the disorder. Therefore, taking into account SA symptoms appears to be particularly relevant in AN. Of note, the cognitive-interpersonal model specifies that avoidance plays an important role, a premise that was not supported in our data. However, these findings should be interpreted with caution as, unlike our other outcome variables, we had over 10% of missing data for social avoidance. An NA of social anxiety and ED symptoms (79.2% AN, mean age = 30 years in the clinical ED sample) found that fears specifically related to social eating and drinking were central [73]. As we used subscale scores rather than single items for the LSAS, we did not examine the role of SA specific to eating and drinking. At the same time, our results are in line with the literature stating that SA is an important ED symptom, particularly during adolescence, a period characterized by increased peer judgment and social challenges.

Interestingly, BMI was not at all connected to the network and was therefore removed from the final network and only included descriptively. One possible explanation is that in this sample of inpatients with severe AN, weight status was relatively homogeneous. At the same time, our finding that BMI is not central to AN is in line with previous literature. One NA of a transdiagnostic mixed adult and adolescent sample found that for all diagnostic subgroups (AN, BN, and BED), BMI had the lowest centrality strength index [40]. In a sample of adolescents (<18 years old) with AN, BMI had the lowest strength centrality as well [41,42]. While BMI is an important indication of the threat to a patient’s life in a hospital context, in terms of theoretical conceptualization, BMI does not appear to be central in the overall symptom constellation and diagnostic criteria of ED. Our results are in line with evidence that points to the centrality of other symptoms over and above BMI.

### 4.2. Bridge Nodes

When performing bridge analysis, three clusters were detected: (1) core ED symptoms, (2) negative affectivity symptoms, and (3) cognitive and socio-affective symptoms. Of note, cognitive and socio-affective symptoms were grouped together in the same cluster. Upon visual inspection of the network, Family Dynamic and Parental Expectations and Concerns connected more strongly with Alexithymia, while perfectionism outcomes (excluding Doubt about Actions) connected more strongly with each other, mirroring the cognitive and interpersonal factors in the model.

The strongest bridge connection from the cognitive and socio-affective cluster to the ED cluster was between Concern over Mistakes and Overvaluation of Weight and Shape. For individuals with EDs, success in achieving weight and shape goals can define most, if not all, of their self-worth. Therefore, the strong association of Overvaluation of Weight and Shape with Concern over Mistakes may reflect the fear of “failing” in the adherence to strict dietary rules to maintain or achieve a certain weight or shape goal. As posited in the cognitive-interpersonal model, people with AN have extremely perfectionistic standards in regard to food consumption and adhere to self-imposed rules rigidly to avoid failure and strong negative emotion [5,6]. However, our results also support the cognitive-behavioral model, which proposes that perfectionistic concerns are a maintaining process tied to the overvaluation of weight and shape [7]. It appears that both models bring forward important arguments about the development and maintenance of ED. Unfortunately, although cognitive flexibility is a key component of the cognitive-interpersonal model, we were unable to examine it more closely. Further research is needed on the core cognitive deficits (i.e., flexibility and central coherence) theorized by the cognitive-interpersonal model before drawing conclusions.

The node with the highest bridge centrality connecting the cognitive and socio-affective cluster to negative affectivity was Doubt about Actions. Bridge nodes are suspected to be key nodes that spread activation from one disorder to another [31]. This result indicates that people with AN who have particularly high levels of this facet of perfectionism may be at a greater risk for developing comorbid depression and anxiety. Interestingly, in our results, Doubt about Actions was not directly connected to core ED symptoms. One NA in a sample of undergraduate students with clinically significant levels of ED and/or OCD symptoms found that Doubt about Actions was an important bridge node between ED and OCD symptoms [74]. Future research should compare people with AN who have comorbid depression, anxiety, and/or OCD with those who do not in order to further examine Perfectionism and Doubt about Actions and the risk of comorbidity.

Depression was a bridge node between negative affectivity and ED symptoms. Specifically, Depression connected most strongly to Preoccupation with Eating. This connection is unsurprising, considering the association between rumination and depression, a link that has also been found in individuals with AN [75]. While no network study has used both the EDEQ subscales and the HAD to assess ED symptoms and depression, respectively, one NA of adolescents with AN found that depression (measured with the Children’s Depression Inventory) was a central node in a network including several general psychopathology nodes and core ED symptoms measured by the Eating Disorder Inventory (EDI) [43]. While the EDI does not have a subscale that maps onto EDEQ Eating Preoccupation, EDI Emotional Dysregulation was connected to Depression, hinting at a potential link between depressive symptoms and maladaptive emotion regulation with food. Future research should examine the role of maladaptive emotion regulation and its potential association with Eating Preoccupation and Depression.

### 4.3. Strengths and Limitations

Our study is the first study to test the cognitive-interpersonal model of AN using NA. This study contributes to the growing body of work using NA to not only examine core ED symptoms, but also their complex relationships with cognitive (perfectionism, flexibility), interpersonal (alexithymia, social anxiety, family dynamic), and mood symptoms (depression, general anxiety) (e.g., [14,39,41]). A strength of our analysis is that we defined adolescence in line with recent biological and social evidence that the period of adolescence extends up to age 24 [47], in contrast to most research, which limits studies of adolescents to less than 18 years old despite no concrete evidence that this represents the end of this critical developmental period. In this age group, we examined severe forms of AN in an inpatient sample with an average illness duration of three years. Investigating central symptoms during this developmental period is critical, as evidence shows that the window for early intervention exists within the first three years of illness, after which treatment outcomes become increasingly poor [76]. Further, understanding the risk factors for severe AN may inform primary prevention of AN, for instance, by aiming to reduce perfectionism in young children and adolescents.

This study is not without limitations. First, as data are cross-sectional, insight into the temporal relationship between symptoms cannot be determined. Second, as previously mentioned, one hospital center failed to collect data for the LSAS Avoidance subscale, potentially affecting results. Further, we used only self-report data, which may not be fully representative; for example, when describing family dynamics, it is the patient’s perception that is measured and not objective dysfunction. We were also unable to conduct an analysis comparing AN subtypes to test the hypothesis posited by the cognitive-interpersonal model that AN-R may have a distinct phenotype. In addition, as our study investigates symptoms in adolescent inpatients, we are not able to draw generalized conclusions regarding anorexia in young children or adults, nor in less severe or subclinical cases. Finally, a small proportion of our participants (*n* = 9) reported a maximum BMI in the overweight/obese range, indicating that our sample may have included participants with atypical AN, who may have different cognitive and clinical profiles compared to “typical” AN. Unfortunately, we lacked statistical power and the data on the medical history of body mass dynamics to analyze this issue separately. At the same time, as this group of patients was small, it is unlikely that they significantly impacted our overall results.

Of note, we had originally planned to include performance on a cognitive flexibility task; however, the task performance variable was not connected to any other node in the network. This is an important limitation to note, as cognitive flexibility is an important component of the cognitive-interpersonal model. As previously mentioned, meta-analytic evidence shows that cognitive flexibility deficits may not be as present in adolescents compared to adults [19,20], which is a potential explanation for why this variable was not connected to the rest of the network. In addition, we did not include an outcome measuring central coherence, which is another important aspect of the cognitive-interpersonal model. Future research should examine cognitive flexibility and central coherence with both cognitive tests and self-report questionnaires, such as the D-Flex [77], to investigate potential deficits more holistically in adolescents. It is important to note that comparing our results to other research is difficult due to our classification of the adolescent age group. Future research should consider redefining the age limits of adolescence in line with the novel chronological framework proposed by Sawyer and colleagues [47], which will significantly impact the evaluation of adolescents and young adults compared to previous research.

## 5. Conclusions

In conclusion, our study provides partial support for the cognitive-interpersonal model while also supporting certain premises put forward by the transdiagnostic cognitive-behavioral model. The most central symptom in our network was related to perfectionism, specifically, Concern over Mistakes. Our hypothesis that Restraint would be a central ED symptom was not supported. Instead, Eating Preoccupation emerged as central, suggesting that the cognitive consequences of dietary restraint (i.e., constant preoccupation with food) play an important role in AN. Further, in line with previous literature, Overvaluation of Weight and Shape was also central to the network. The high centrality of Concern over Mistakes and social anxiety supports the theory that both cognitive and interpersonal difficulties contribute to AN, particularly in adolescence. Our results suggest a potential compatibility between the cognitive-behavioral theory and the cognitive-interpersonal model.

## Figures and Tables

**Figure 1 children-10-00730-f001:**
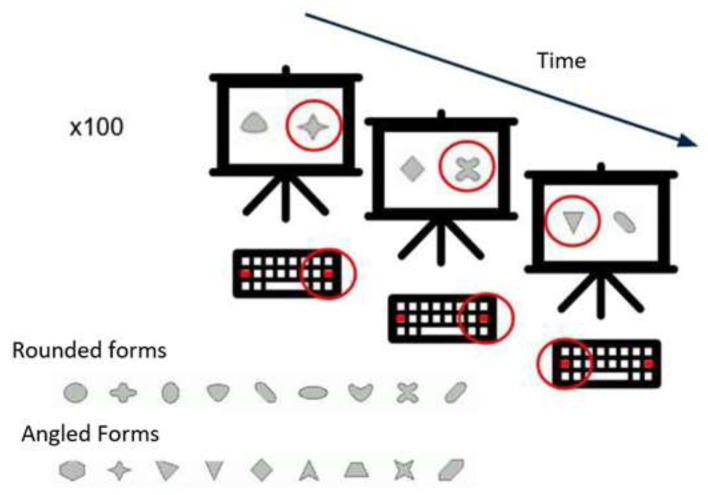
Illustration of the TAP 2.1 Flexibility subtest. The red circle indicates the correct response.

**Figure 2 children-10-00730-f002:**
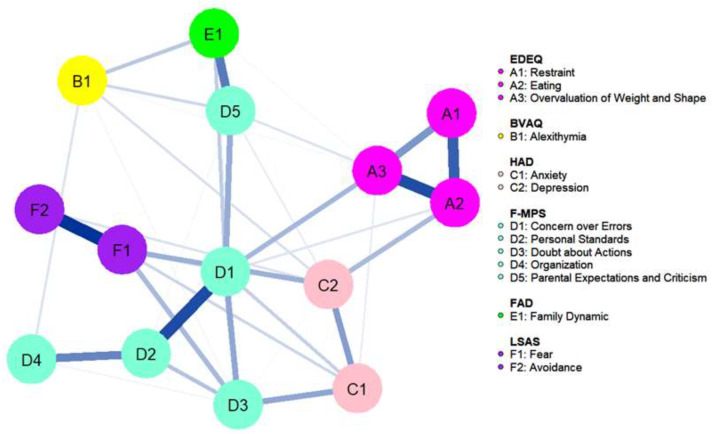
Network plot. Nodes are represented as circles, color coded for each questionnaire (EDEQ: Eating Disorder Examination Questionnaire, BVAQ: Bermond–Vorst Alexithymia Questionnaire, HAD: Hospital Anxiety Depression Scale, FMPS: Frost Multidimensional Perfectionism Scale, FAD: Family Assessment Device, LSAS: Leibowitz Social Anxiety Scale). Blue lines between nodes represent a positive association; thicker lines correspond to stronger associations.

**Figure 3 children-10-00730-f003:**
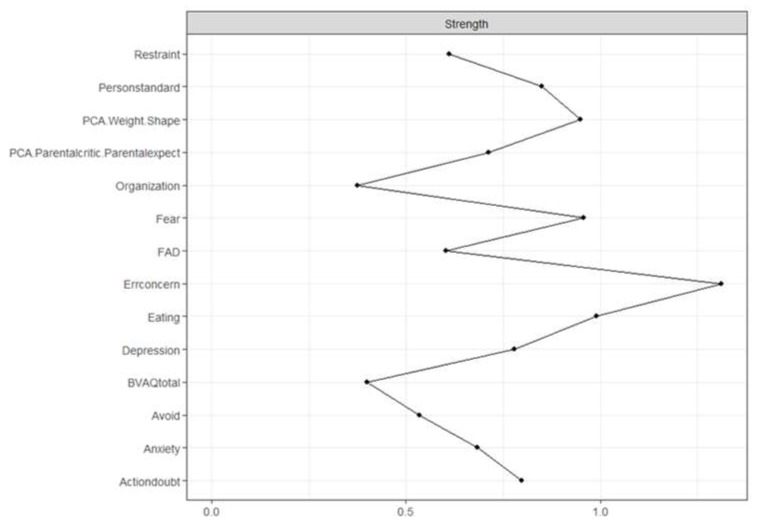
Errconcern: Concern over Mistakes, Actiondoubt: Doubt about Actions, PCA.WeightShape: Overvaluation of Weight and Shape, PersonStandard: Personal Standards, BVAQ total: Alexithymia total score, PCA.Parentcritic.ParentExpect: Parental Criticism and Expectations, FAD: Family Assessment Device.

**Figure 4 children-10-00730-f004:**
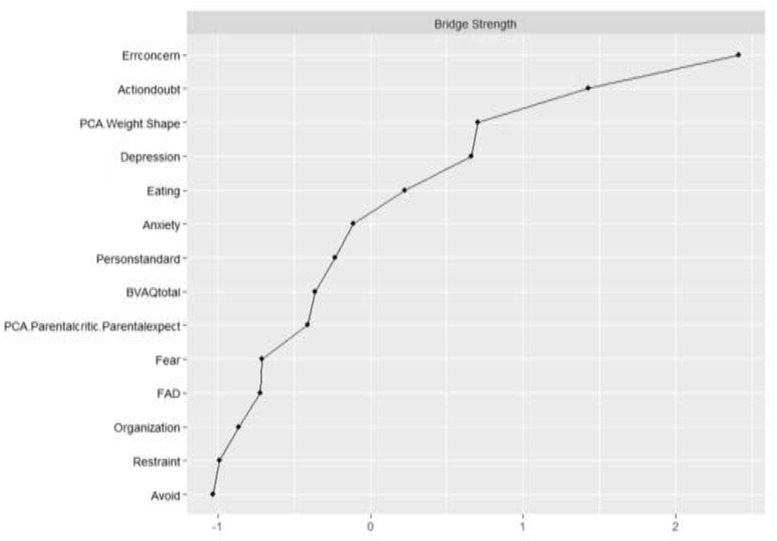
Bridge strength plot. Errconcern: Concern over Mistakes, Actiondoubt: Doubt about Actions, PCA.WeightShape: Overvaluation of Weight and Shape, PersonStandard: Personal Standards, BVAQ total: Alexithymia total score, PCA.Parentcritic.ParentExpect: Parental Criticism and Expectations, FAD: Family Assessment Device.

**Figure 5 children-10-00730-f005:**
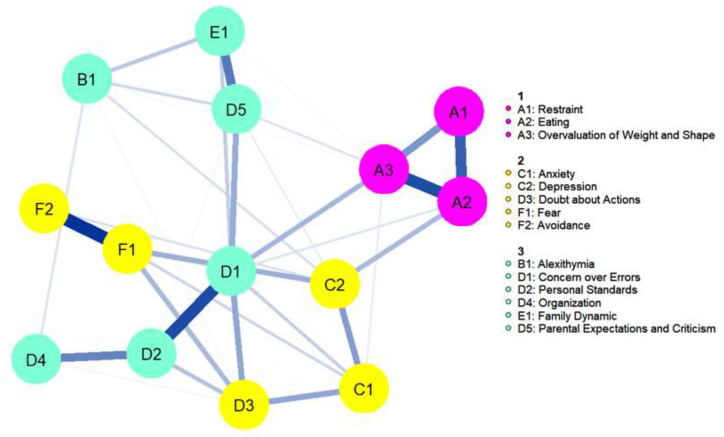
Network plot with bridge clusters. Nodes are represented as circles, color coded for each community (1: core eating disorder symptoms, 2: negative affectivity, 3: cognitive and socio-affective symptoms). Blue lines between nodes represent a positive association; thicker lines correspond to stronger associations.

**Table 1 children-10-00730-t001:** Descriptive statistics.

	N	Mean	Median	SD	Min	Max
Age of Onset	137	15.3	15.0	3.4	6.0	23.0
Age at Admission	145	18.2	17.7	3.1	13.2	24.5
Duration Of Illness (Years)	137	2.9	1.9	2.5	0.1	12.1
Previous Hospitalizations	129	2.9	2.0	4.2	0.0	29.0
BMI At Evaluation	145	14.3	14.1	1.6	10.3	18.9
Minimum BMI	145	13.2	13.3	1.7	7.6	18.5
Maximum BMI	143	19.9	19.5	3.2	13.0	30.5
EDEQ						
Restraint	144	2.8	2.8	1.8	0.0	6.0
Eating	145	2.8	2.8	1.4	0.0	6.0
Shape	144	3.9	4.0	1.4	0.0	6.0
	144	3.4	3.4	1.4	0.0	6.0
Anxiety	145	12.0	12.0	4.4	2.0	21.
Depression	145	8.9	9.0	4.4	0.0	18.0
Alexithymia	142	56.8	57.0	9.5	34.0	92.0
Concern over Mistakes	142	30.6	32.0	8.6	9.0	45.0
Personal Standards	143	25.6	26.0	5.3	11.0	35.0
Parental Expectations	143	11.3	11.0	5.0	5.0	24.0
Parental Criticism	143	9.2	9.0	3.5	4.0	20.0
Doubt about Actions	143	13.4	14.0	3.7	4.0	20.0
Organization	142	24.9	25.0	3.9	11.0	30.0
Fear	140	30.9	29.5	16.1	1.0	68.0
Avoidance	122	14.4	13.0	10.3	0.0	41.0
Family Global Functioning	141	2.1	2.1	0.6	1.0	3.8
FLEX Median (Seconds)	145	729.4	691.0	172.8	417.0	1295.0

Note: BMI = body mass index; EDEQ = Eating Disorder Examination Questionnaire; FLEX = Test of Attention Performance Cognitive Flexibility Subtest.

**Table 2 children-10-00730-t002:** Descriptive statistics.

Node Name	Strength	Bridge Strength	Predictability
Restraint	0.61	0.01	0.48
Eating	0.99	0.22	0.64
Overvaluation of Shape and Weight	0.95	0.30	0.60
Alexithymia	0.40	0.12	0.15
Anxiety	0.68	0.16	0.40
Depression	0.78	0.29	0.39
Concern over Mistakes	1.31	0.60	0.66
Personal Standards	0.85	0.14	0.56
Doubt about Actions	0.80	0.42	0.44
Organization	0.37	0.03	0.21
Parental Expectations and Criticism	0.71	0.11	0.35
Family Dynamic	0.60	0.05	0.30
Social Fear	0.96	0.06	0.53
Social Avoidance	0.53	0.00	0.38

## Data Availability

The data presented in this study are available upon reasonable request from the corresponding author. The data are not publicly available due to specifications from the study sponsor.

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
