# Peer review of "Cognitive and Interpersonal Factors in Adolescent Inpatients with Anorexia Nervosa: A Network Analysis"

_children, 2023, doi:10.3390/children10040730_

Round 1

Reviewer 1 Report

This is a quality article, based on interesting research. The article is well-written and talks about a very important topic, especially in pediatric medicine today.  

Only minor corrections to the English language and pronunciation are needed, along with a few additional corrections as listed below:

Line 66: The subsection is marked with the wrong number

Line 193 & 447:

In Materials and Methods, subsection Study Population, the characteristics of the subjects, inclusion and exclusion criteria are described, but the term ‘severe AN’ is not mentioned. This characteristic of the subjects is mentioned in the title and then in the article for the first time only in the discussion, line 447; ‘The aim of this study was to evaluate the cognitive-interpersonal model in a sample of adolescents with severe AN’

How did the authors define, according to which criteria, that it is a severe form of the disease, severe AN? This is the necessary minimum that must be stated in the Materials and Methods chapter.

Line 197: 

Are other organic diseases excluded in all patients, especially celiac disease and brain tumors? Namely, these are diseases that represent an important differential diagnostic problem since, mainly unrecognized and untreated, they can affect behavior and cognition.

Line 202:

Where were the adolescents hospitalized, whether they were in pediatric wards or child and adolescent psychiatry ward?

Line 205:

These are old data, why are they being sent to publication only now, after more than 10 years have passed?

Line 210-211:

Materials and methods list low BMI and/or sudden and rapid weight loss as inclusion criteria: ‘Inclusion criteria for the current study were as follows: being hospitalized for AN, admission BMI<15 and/or sudden and rapid weight loss, consent for participation in the study, and affiliation with the French Social Security health coverage system.’

Here, in Measures, only the absolute value of BMI is reported. 

So what was ultimately used, and what were the inclusion criteria?

If only the absolute value of BMI is taken in one slice of time, there is a risk of missing adolescents with a sudden loss of body mass who at the time of examination are still of preserved BMI but may have all the complications of starvation as well as severely malnourished adolescents with AN. It is a group of so-called atypical anorexia nervosa.

Line 286:

As in the previous suggestion, the question arises whether the researchers collected data from the medical history of body mass dynamics. It is not enough only to know about the current BMI, but also the dynamics due to starvation, which is the main risk factor for the development of a severe clinical presentation of AN with numerous metabolic complications and an impact on cognition. Namely, this group of adolescents, who were obviously overweight before the disease, can have even more severe metabolic complications than those adolescents who, before the onset of the disease, were in the lower limits of normal BMI, ‘healthy thin’.

Line 375:

It is not stated here whether this is a group of patients who were diagnosed with AN for the first time or if there are patients who relapsed among them, or if they were previously treated in another institution.

Line 381:

The duration of the disease is very long, 2.9 years! So, I recommend stating clearly whether this is the first hospitalization for AN or not.

Line 447:

As previously mentioned, it is important to clarify the criteria by which patients are categorized as severe AN.

Line 526-534:

Here the importance of starvation is nicely elaborated in accordance with my previous suggestions. It would be interesting to see if the stratification of patients according to the rate of weight loss has an impact on the obtained results.

I recommend the reference: Garber AK, Cheng J, Accurso EC et al. Weight Loss and Illness Severity in Adolescents With Atypical Anorexia Nervosa. Pediatrics. 2019 Dec;144(6). doi: 10.1542/peds.2019-2339.

Line 583:

The strengths and limitations of this research are nicely described.

Line 611-612:

I recommend additionally stressing the difficult comparison of the results with other research due to the different classification of age groups and the recommended new chronological framework of adolescence, with which I fully agree. This ‘pioneering’ approach, redefining the age limits of adolescence, will have a significant impact on the evaluation of data compared to previous research.

Reviewer 2 Report

Dear Authors,

The article Cognitive and Interpersonal Factors in Adolescents with Severe Anorexia Nervosa: A Network Analysis by Chantal P. Delaquis et al. represents an important and valuable position on issues related to AN.

I read it with great interest, but raised several concerns:

#1. The introduction is very comprehensive, but the authors should emphasize more the current state of knowledge and what their study will contribute to science. 

#2. Please add the hypothesis of this study in the introduction.

The chapter material and methods is described very well and is not objectionable. Congratulations on the large study group.

The results, for the most part, should be considered very interesting.

The discussion has been properly guided with reference to relevant literature. 

#3 Please add limitations in more detail, especially regarding the facility where the study was conducted and the inability to draw generalized conclusions. 

#4 The authors should formulate specific conclusions that can be made on the basis of the study conducted, the others can be found in the subsection further research directions.

Congratulations on a very good article. Your research is well structured, insightful and makes a valuable contribution to the field. I can see that you have put a lot of effort into your work, which is evident in the high quality of your writing and analysis. Overall, I believe that your article has the potential to make a significant impact on your field and I am confident that it will be well received by the scientific community.

Greetings 

Reviewer 3 Report

Thank you for the possibility to review this manuscript. This manuscript aimed to evaluate the cognitive–interpersonal model in a sample of adolescents with severe AN, specifically the complex relationships between core ED symptoms and cognitive, interpersonal, and mood symptoms. The manuscript is well–written and easy to read, and the results are presented clearly and with great attention.

There are my minimal comments:

Ethical considerations are missing.

All referencing should be presented in one bracket, i.e, [45, 46], not [45], [46].

An original reference for EDEQ should be inserted.

Cronbach alphas of separate subscales should be presented in the Methods (instruments section).

Round 2

Reviewer 2 Report

Dear Authors,

Thank you for considering my comments. Once again, congratulations on an excellent article and continued fruitful scientific research. 

Greetings!